# A Methodology for Applying Skew in an Automotive Interior Permanent Magnet Rotor for Robust Electromagnetic and Noise, Vibration and Harshness Performance

Thomas Cawkwell [1], Ahmed Haris [2], Juan Manuel Gonzalez [3], Leon Kevin Rodrigues [4],* and Vladimir Shirokov [5]

1    Independent Researcher, London KT6 4DA, UK
2    Arrival Co., Ltd., London W14 8TS, UK; ahmedharis@hotmail.co.uk
3    Independent Researcher, Guildford GU4 7XT, UK; juanmg85@yahoo.com
4    Textron Specialized Vehicles, Ipswich, Suffolk IP3 9TT, UK
5    Independent Researcher, London SW16 4QE, UK
*    Correspondence: leon.k.rodrigues@gmail.com

**Abstract:** Interior permanent magnet (IPM) motors in traction applications often employ discrete rotor skewing constructions to reduce torsional excitations and back-EMF harmonics. Although skewing is very effective in reducing cogging torque, the impact on torque ripple is not well understood and can vary significantly over the operating envelope of a motor. Skewing also leads to the creation of a non-zero axial force that may compromise the bearing life if not considered. This paper introduces a holistic methodology for analyzing the effect of skewing, aiming to minimize torsional excitations, axial forces and back-EMF harmonics whilst mitigating the impact on performance and costs. Firstly, analytical models are employed for calculating cogging torque, torque ripple and axial forces. Then, 2D and 3D finite element analysis are used to incorporate the influence of non-linear material behavior. A detailed structural model of the powertrain is employed to calculate the radiated noise and identify key areas allowing a motor designer to reduce noise, vibration and harshness (NVH). A meticulous selection process for the skewing angle, the number of skew stacks and the orientation of skew stacks is developed, giving particular attention to the effect of the selected pattern on NVH in both forward and reverse rotating directions.

**Keywords:** interior permanent magnet (IPM) motor; skewing; NVH; torque ripple; axial force

## 1. Introduction

The interaction of the space harmonics in the stator and rotor magnetomotive forces (MMFs) of electric motors leads to the creation of non-fundamental components in the output torque waveform. This phenomenon is commonly referred to as torque ripple, and in IPM machines, it is often aggravated by the selection of the location and orientation of the magnets in the rotor lamination that are meant to benefit other targets like high reluctance torque or low magnet loss at high speed.

If no design countermeasures are applied, the peak-to-peak amplitude of the torque ripple signal can be as high as 40–50% of the average torque. While this is undesirable in a wide range of applications, it is particularly detrimental in automotive traction motors since it can excite vibration modes and create acoustic noise that can cross the admissible thresholds and reduce passenger comfort. The study performed in [1] analyzes the contribution to NVH in different frequency ranges from two different sources: torque ripple and radial force. Several approaches to minimizing motor noise in IPMs have been explored in several research papers, including the following:

- Pole–slot number variation is described in [2], where with the help of the Maxwell stress tensor and equivalent magnetizing current method, the authors study two different motors with similar performances but different pole–slot combinations.

- Two-dimensional geometry optimization is applied with the aim of reducing the mechanical sources of torque ripple and electromagnetic forces. This methodology, outlined in [3], also manipulates structural resonances to augment the overall stiffness characteristics.
- Utilization of asymmetric slot design allows for non-symmetrical magnetic reluctance of the rotor around the d-axis. As illustrated in reference [4], this approach can be advantageous under certain working conditions but poses manufacturing challenges for automotive applications.
- Rotor notching, as investigated in [5,6], exhibits a potential reduction in torque ripple, contributing to a diminished NVH response. The influence of notch dimension, location, and quantity was examined in [5].
- Skewing of the rotor or stator is a widely adopted method to minimize NVH. However, determining the optimal skew parameters is challenging and may result in suboptimal designs. The authors of [7] analyze rotor skewing for short-length and highly saturated machines, emphasizing the importance of the fringe effect and challenging the validity of 2D multi-slice modeling. The authors of [8] investigate the effects of skewing for different magnet shapes and pole–slot combinations yet neglect 3D effects such as axial force and its relation to the order of rotor stacks also referred to as slices. Reference [9] focuses on structural analysis coupled with electromagnetic FEA to enhance excitation force harmonics, aiming to reduce motor core deformations that lead to vibration and noise.

This study introduces a methodology for employing skew in interior permanent magnets for automotive applications with the aim of minimizing the radiated noise whilst reducing the impact on average torque and manufacturing costs. In this study, a typical automotive traction IPM motor was utilized whose specifications are presented in Table 1.

**Table 1.** Traction motor specification.

| Parameter | Value | Unit |
| --- | --- | --- |
| Number of slots—Q | 72 | - |
| Number of poles—2p | 12 | - |
| Motor torque | 400 | Nm |
| Motor maximum speed | 8700 | rpm |
| Maximum Voltage | 400 | V |
| Maximum Power | 155 | kW |
| Maximum Current | 400 | A |

The skew implementation can be obtained in two different ways. If applied to the stator, the process involves a continuous rotation of laminations, as depicted in Figure 1a. On the other hand, when skewing the rotor, while a continuous pattern is feasible for wound and squirrel cage applications, this is not practical in permanent magnet rotors, wherein skew is implemented through a series of stepped sections to maintain manufacturing simplicity. It is crucial to select the appropriate number of sections for skewing and their orientation. Typically, these sections can be arranged in a linear or V-skew configuration. The V-skew, a proven method to reduce axial forces in linear skewing [10], helps minimize these forces, which, although small, can contribute to increased NVH. However, the drawback of the V-skew lies in its need for a larger number of rotor sections, resulting in an increase in manufacturing costs and complexity.

Stacks of different lengths as studied in [11], however appealing from an optimization point of view, imply magnets of different lengths, adding cost and complexity. To simplify the manufacturing process, a partial V-skew where the orientation of the linear skew segments is re-arranged to mimic a V-shape skew without increasing the number of segments or zig-zag orientation was suggested by Blum et.al. [12] and is shown in the example in Figure 1b. This method is only applicable for more than two skew steps, but due to the non-symmetrical arrangement of the skewed steps, the partial V-skew does not fully cancel

out axial forces. This paper analyzes all the possible options of the partial V-skew for a three- and four-step skew.

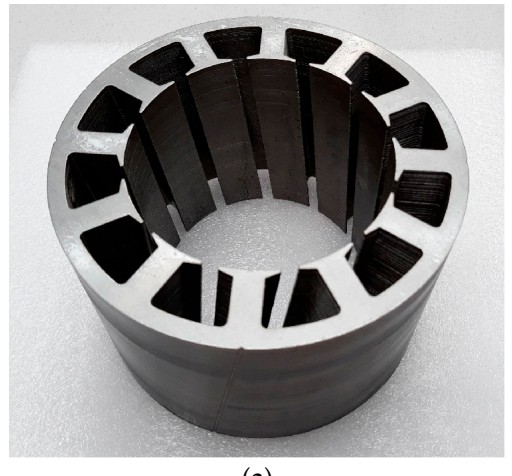 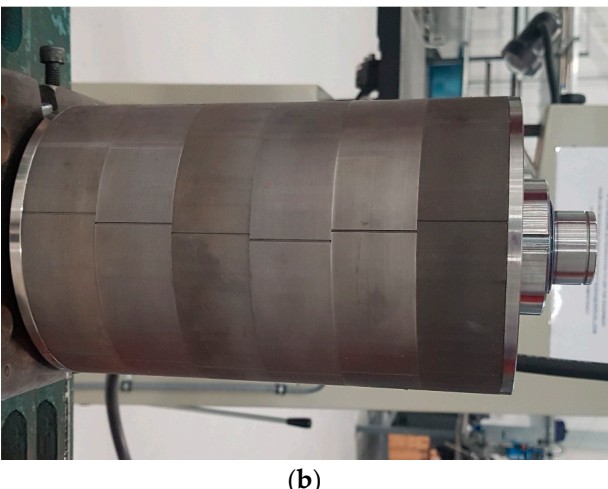

(**a**)                                    (**b**)

**Figure 1.** Skewing examples: (**a**) stator skewing (continuous); (**b**) IPM rotor with partial V-skew (discrete).

The paper introduces a novel methodology utilizing tools of increasing accuracy and computational complexity to analyze the impact of the skewing parameters on torque ripple, cogging torque and axial force and presents a well-defined process for applying this method in a real motor design project. The subsequent sections are also structured in increasing complexity, as follows:

In Section 2, the analytical models for the main excitations, including torque ripple, cogging torque and axial forces, are presented. These calculations are meant to provide an initial estimation of the order of magnitude for each one of them, offering a preliminary understanding of the impact of skewing on performance.

In Section 3, a 2D finite element model is employed to obtain the initial un-skewed or baseline case, and then different skewing options are applied to it. These are obtained by emulating the mechanical skewing angle between stacks by means of a different phase current angle for each.

In Section 4, a 3D finite element model is used to assess the different combinations of the number of stacks, and their assembly order is studied with particular interest in the effect on the resultant axial force generated.

A detailed structural model is used in Section 5 to analyze the impact of skewing on the total radiated noise using a structural model, wherein the average surface velocity of the housing is computed across different harmonics, both in the presence and absence of skew.

Finally, the impact of skewing in the manufacturing process is evaluated in Section 6, and an NVH optimization workflow using all the tools described in this paper is shown in Section 7.

## 2. Analytical Modeling of Torque Ripple, Cogging Torque and Axial Forces

### 2.1. Torque Ripple

Torque ripple in electrical motors can be caused by several reasons such as rotor and stator field harmonics, non-sinusoidal current excitation and cogging. The torque equation of an IPM motor is given by

$$T = \frac{3}{2} p i_a \psi_a \cos(\beta) \tag{1}$$

where $p$ is the number of pole pairs, $\psi_a$ is the phase flux linkage, $i_a$ the phase current and $\beta$ is the phase angle relative to the motor's direct axis. In the case where the currents are sinusoidal

$$i_a = -\sqrt{2}I_a \sin(\omega t + \beta) \tag{2}$$

the source of the torque ripple is attributed to harmonics in the flux linkages and can be represented as

$$\psi_a = \sum_{k=1,3,5,..}^{\infty} \psi_k \cos(k\omega t + \alpha_k) \tag{3}$$

where $k$ is the harmonics number of the flux linkage, $\psi_k$ is the harmonic coefficient and $\alpha_k$ is the phase angle of the harmonic. In balanced three-phase motors, all third multiple harmonics have a 0 coefficient. For the purpose of this analysis, only the fundamental and first four harmonics are utilized:

$$\psi_a = \psi_1\cos(\omega t) + \psi_5\cos(5\omega t + \alpha_5) + \psi_7\cos(5\omega t + \alpha_7) \\ + \psi_{11}\cos(11\omega t + \alpha_{11}) + \psi_{13}\cos(13\omega t + \alpha_{13}) \tag{4}$$

By using Equations (4) and (2) in (1), we can calculate the torque of the motor as

$$T = \tfrac{3}{2}p\sqrt{2}I_a \left[\psi_1\cos(\beta) - 5\psi_5\cos(6\omega t + \alpha_5 + \beta) + \psi_7\cos(6\omega t + \alpha_7 - \beta) \right. \\ \left. - \psi_{11}\cos(12\omega t + \alpha_{11} + \beta) + \psi_{13}\cos(12\omega t + \alpha_{13} - \beta)\right] \tag{5}$$

The mechanical frequency of the torque ripple harmonics can be extracted from Equation (5) and represented as

$$f_r = 2npm f_m \tag{6}$$

where $m$ is the number of phases, $f_m$ is the fundamental mechanical frequency and $n$ is the harmonic number (1, 2, 3, . . .).

### 2.2. Cogging Torque

Cogging torque is a component of torque ripple which is caused by the change in co-energy in the airgap as the rotor pole and stator teeth move from alignment to misalignment or vice versa in the absence of the stator current. While every real electrical rotating machine, to some extent, exhibits torque ripple, cogging torque is a unique feature found only in permanent magnet machine types. The cogging harmonic frequencies $f_{cog}$ due to the rotor and stator slotting harmonics are a function of the least common multiple (LCM) of stator slots ($Q$) and rotor poles ($2p$):

$$N_{cog} = \text{LCM}(Q, 2p) \tag{7}$$

$$f_{cog} = nN_{cog}f_m \tag{8}$$

The cogging torque with rotor position can be estimated using the airgap permeance function $G(\theta)$ and airgap flux density $B(\theta, \alpha)$ of the rotor using [13]

$$T(\theta) = \frac{\pi L_s}{4\mu_o}\left(r_{stat}^2 - r_{rot}^2\right) \sum_{n=1}^{\infty} nN_{cog}G_{nN_{cog}}B_{nN_{cog}}\sin(nN_{cog}\theta) \tag{9}$$

where $L_s$ is the motor stack length, $r_{stat}$ and $r_{rot}$ are the radii of the stator and rotor, and $G_{nN_{cog}}$ and $B_{nN_{cog}}$ are the Fourier coefficients of the airgap permeance function and flux density. A method for calculating the Fourier coefficients analytically was proposed by [14] and given as

$$G_{nN_{cog}} = \frac{2Q}{n\pi N_{cog}}\sin\left(\frac{nN_{cog}b_o}{2}\right) \tag{10}$$

$$B_{nN_{cog}} = \frac{2(2p)}{n\pi N_{cog}}B_s^2\sin\left(\frac{nN_{cog}\pi\gamma_p}{2}\right) \tag{11}$$

where $b_o$ is the slot opening of the stator, $B_s$ is the peak airgap flux density of a slot less stator equivalent and $\gamma_p$ is the pole-arc-to-pole-pitch ratio.

The most significant cogging and torque ripple harmonics calculated using (6) and (8) for the 72-slot 12-pole motor in Table 1 are outlined in Table 2. The two main torsional harmonics are the 36th and 72nd orders relative to the mechanical frequency. The higher-order harmonics have a lower magnitude which leads to a lower response and hence have not been considered.

**Table 2.** Torsional harmonic excitation order and type.

| Order | Harmonic | Excitation |
|---|---|---|
| 1 | 36 | Torque ripple |
| 2 | 72 | Cogging + Torque ripple |
| 3 | 108 | Torque ripple |
| 4 | 144 | Cogging + Torque ripple |

### 2.3. Axial Force

A preliminary mathematical approximation of the axial force arising from a skewed rotor can be formulated by applying the Lorentz force to a current-carrying conductor $L$ exposed to a magnetic induction field $B$. In the simplest case of a non-skewed rotor, $B$ represents the average value in the airgap and is solely oriented in the radial and tangential directions. This is illustrated in Figure 2a, where two sets of IPM magnets, axially displaced, occupy the same angular position. The stator current $i$ flows in the axial direction but is omitted in the scheme for simplicity. This non-skewed configuration yields a force given by

$$\overline{|F|} = i \cdot \overline{|L \times B|} = i \cdot L \cdot B \cdot \sin\left(\frac{\pi}{2}\right) = i \cdot L \cdot B \tag{12}$$

where $\pi/2$ denotes the orthogonality between the flux density and stator current. As a consequence, the direction of the resultant force has no axial component.

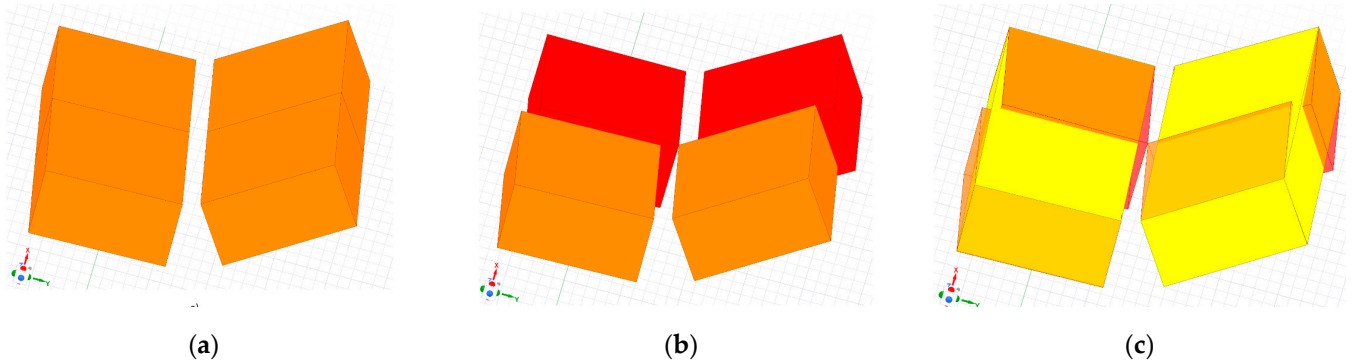

| (a) | (b) | (c) |

**Figure 2.** Magnets from two consecutive axial segments: (**a**) non-skewed consecutive stacks (orange); (**b**) discrete skewing (magnets from stack 1 in orange and stack 2 in red); (**c**) continuous skewing in yellow overlapping case (**b**) for analytical approximation of axial force.

The discrete skewed case shown in Figure 2b, where two sets of axially displaced magnets present an angle between their d-axes, is characteristic of a real IPM rotor. However, this case is complex to model analytically due to the axial discontinuity between stacks. An approximation can be offered by replacing this configuration with one set of magnets of double the length that is continuously skewed by an angle $(\delta_i - \delta_j)$, where $\delta_i$ and $\delta_j$ are the angles of the d-axes of each of the previous sets, respectively measured from the reference line (the d-axis of the non-skewed rotor). This is shown in yellow in Figure 2c, overlapping

the discrete magnets of case 2b. By placing the reference system in the rotor, the Lorentz force can be obtained from the cross-vector operation:

$$\overline{F} = i \cdot \overline{L} \times \overline{B} = i \cdot L \begin{vmatrix} \check{t} & \check{r} & \check{z} \\ \sin(\delta_i - \delta_j) & 0 & \cos(\delta_i - \delta_j) \\ B_t & B_r & 0 \end{vmatrix} \tag{13}$$

where $\check{t}$, $\check{r}$ and $\check{z}$ are the tangential, radial and axial components of the force. In particular, the axial component is given by

$$\overline{F}_z = i \cdot L \cdot B_r \cdot \sin(\delta_i - \delta_j) \tag{14}$$

To calculate the full expression of the axial force, the current in (14) must be replaced by the stator magnetomotive force ($MMF_S$):

$$MMF_S = Q \cdot \frac{I_a}{a} \cdot N_{Cond} \tag{15}$$

where $a$ is the number of parallel branches per phase, $I_a$ is the RMS phase current and $N_{Cond}$ is the number of conductors per slot. Replacing (15) in (14), the expression of the axial force between two skewed stacks is obtained by

$$\overline{F}_z = Q \cdot \frac{I_a}{a} \cdot N_{Cond} \cdot L \cdot B_r \cdot \sin(\delta_i - \delta_j) \tag{16}$$

It is important to note that $L$ is the active length of the two stacks in consideration and should not be mixed with the total active length $L_s$. Additionally, the sign of the angles of each set must be respected.

## 3. Two-Dimensional FEA Modeling

### 3.1. Torque Ripple Excitations without Skew

A 2D transient finite element model of the motor was built using Ansys Motor-CAD. The number of points per mechanical cycle of 1440 was selected to ensure sufficient fidelity was available to capture the 144th harmonic. The cogging torque of the motor was calculated and compared to the analytical model described in Section 2.2 and is shown in Figure 3. Figure 4 shows the comparison of the analytical model of torque ripple from Section 2.1 and FEA calculations for 150 A and 1000 rpm. Both cogging and torque ripple 2D FEA results show very good correlation to the analytical models.

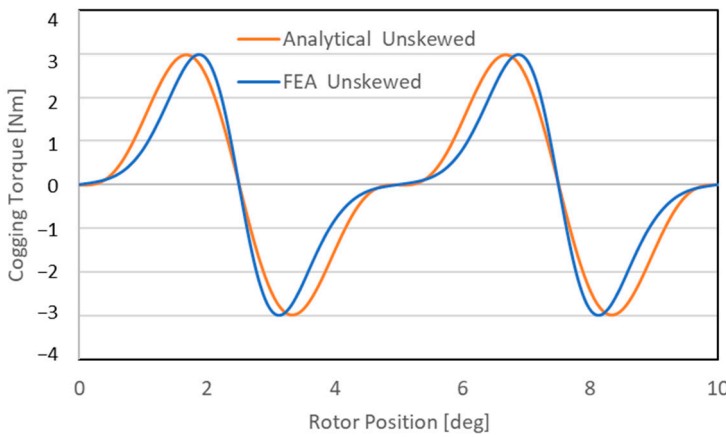

**Figure 3.** Comparison of analytical and FEA models of cogging torque.

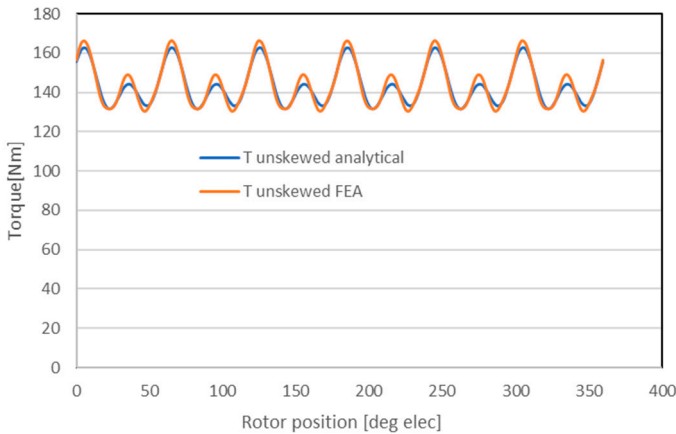

**Figure 4.** Comparison of analytical and FEA models of torque ripple at 150 A and 1000 rpm.

The peak-to-peak torque ripple values of the machine without skew at a selection of operating speeds and torque points were extracted using the FEA model and are shown in Figure 5a. The analysis reveals that peak-to-peak values exhibit notable variation throughout the motor's operational range. These values reach their maximum absolute magnitude within the low-speed and high-torque range. However, when considered as a percentage change, the peak-to-peak values are more pronounced in the high-speed and low-torque range. Harmonic analysis was carried out on the transient torque waveforms to extract the contributions of the individual harmonics and is shown in Figures 5b and 5c for the 36th and 72nd harmonics, respectively, for the same operating points. In this motor design, the 36th harmonic is dominant in the field-weakening region, whereas the 72nd harmonic is dominant at lower speeds or in the maximum torque per ampere (MTPA) region.

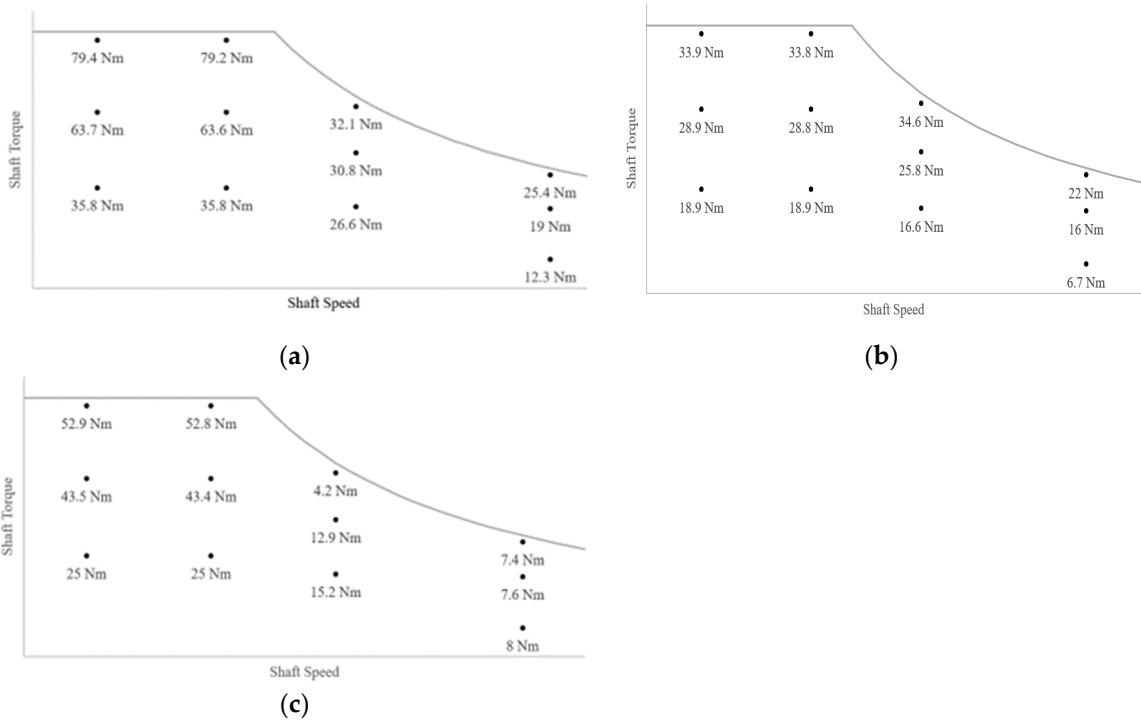

**Figure 5.** (**a**) Total peak-to-peak torque ripple magnitude; (**b**) peak-to-peak torque ripple magnitudes of the 36th harmonic; (**c**) peak-to-peak torque ripple magnitudes of the 72nd harmonic for a motor without skew.

*3.2. Impact of Stator Skew*

The selection of the skew angle typically relates to the wavelength of the space harmonic that needs to be minimized. The impact of a given skew angle $\delta_{sk}$ on the individual harmonics can be defined by the skewing factors $k_n$:

$$k_n = \frac{\sin\left(n\frac{\delta_{sk}}{2}\right)}{n\left(\frac{\delta_{sk}}{2}\right)} \tag{17}$$

The skew factors calculated against the mechanical skewing angle for the harmonics in Table 2 are shown in Figure 6.

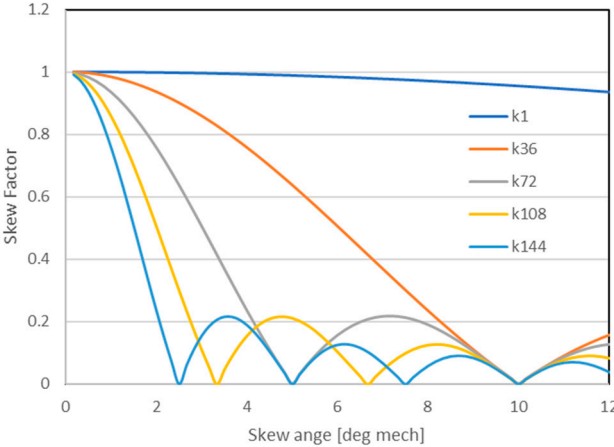

**Figure 6.** Skewing factors for the fundamental torque and main motor harmonics.

The impact of skew using analytical and FEA methods is shown in Figure 7 for a mechanical skew angle of 5°. As expected from the skew factors, the 72nd and 144th harmonics are minimized with this angle, but torque ripple which has components of the 36th and 108th harmonics is not canceled out. To establish the effect on torque ripple for increasing rotor skew angles, the 2D transient FEA simulations described above were repeated at different skew angles across three shaft rotating speeds at phase currents of 150 A and 350 A. Increasing the skew angle also has a negative impact on the peak torque, and the trade-off between skew angle and peak performance is evaluated.

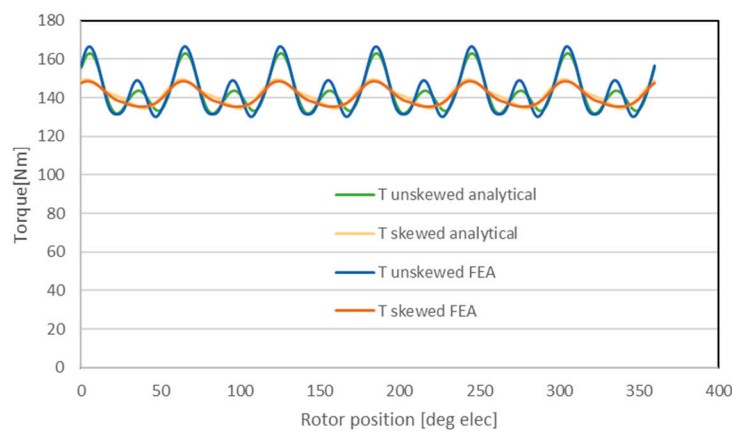

**Figure 7.** Impact of a 5° skew using analytical and FEA models at 150 A and 1000 rpm.

Figure 8 illustrates the correlation between the reduction in total torque ripple and the stator skew angle at 150 A and 350 A, considering various speeds, and its subsequent impact on the peak torque. At lower speeds, a significant reduction in torque ripple can be seen for all current values with the introduction of a skew of between 4° and 6°,

thus indicating that the targeted 72nd harmonic cogging component is dominant at these operating points. Increasing the skew angle to 10° eliminates torque ripple as it minimizes the 36th and 72nd harmonics as well as all their higher-order multiples. Nonetheless, this enhancement is accompanied by a 6.5% decrease in peak torque. Additionally, a stator skew in a hairpin-like design increases manufacturing complexity and hence the cost of the stack and winding.

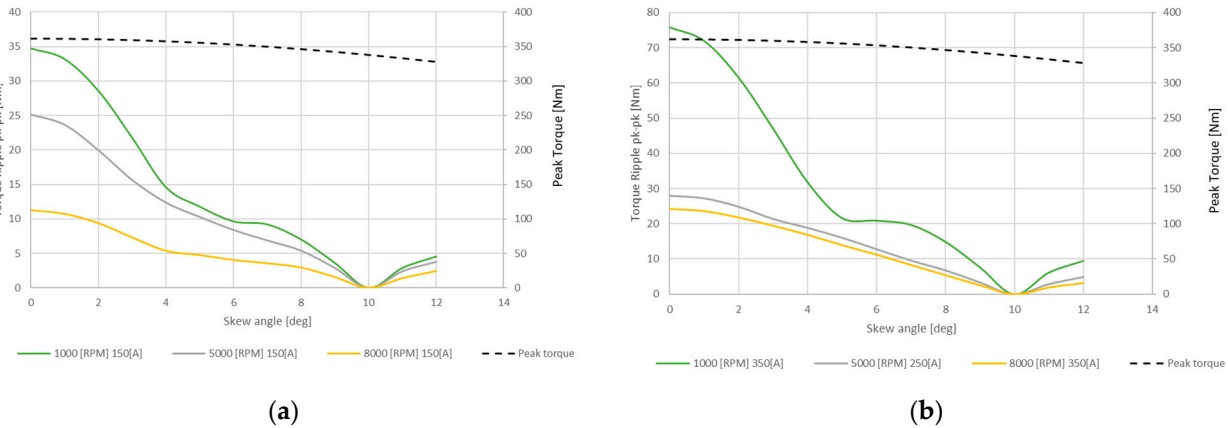

(**a**)                                                                                                (**b**)

**Figure 8.** Impact of skew angles on the normalized torque ripple and peak torque using stator skew at different rotor speeds: (**a**) 150 A; (**b**) 350 A.

### 3.3. Impact of Rotor Skew

When implementing rotor skew with an effective skew angle of $\delta_{sk}$, the angular displacements $\alpha$ between consequent segments of the rotor section are given by

$$\alpha = \frac{\delta_{sk}}{\epsilon} \tag{18}$$

where $\epsilon$ is the number of skew steps. This formula assumes the rotor is skewed into equal lengths. In order to simulate the stepped skew section, a multi-slice 2D FEA simulation that was shown to be a reasonable approximation of a full 3D simulation has been used [15].

Figure 9 shows the impact of skew angle on torque ripple and peak torque employing a rotor skew with $\epsilon = 4$ steps. It shows that in the MTPA region, a similar behavior to the stator skew is achieved with an optimum skew angle between 5° and 6°. At higher speeds where the motor is operating in field weakening, the 150 A current case shows a significant reduction in torque ripple, but at 350 A, the effect of skewing has a marginal effect. More importantly, a complete cancellation of torque ripple is not possible at any skew angle. Unlike stator skew, the rotor skew sections produce different peak torque and ripple as the individual sections are exposed to different phase advance angles. The authors of [11] showed that modifying the skew angles and lengths of the individual sections can be exploited to further reduce cogging torque. It should be noted that using sub-stacks with different stack lengths also increases manufacturing complexity as different magnet sizes and stacks must be sourced for the same rotor, increasing the part count and process costs.

Harmonic analysis of the transient torque ripple waveforms was carried out, and the impact of the rotor skew angle on the individual harmonic magnitudes is shown in Figure 10 for the three speeds at 350 A. The results show that the optimum skew angle for the 72nd harmonic varies significantly with rotor speed. At 1000 rpm, the 72nd harmonic is minimized with the typical skew angle of 5°, whereas at 8000 rpm, a slightly higher 6° skew shows the best results. At around 5000 rpm, the 72nd harmonic appears to be naturally minimized by the phase angle of the current, and therefore, the skew angle increases the 72nd harmonic ripple. At 8000 rpm, minimum torque ripple is achieved with a skew angle of 8°.

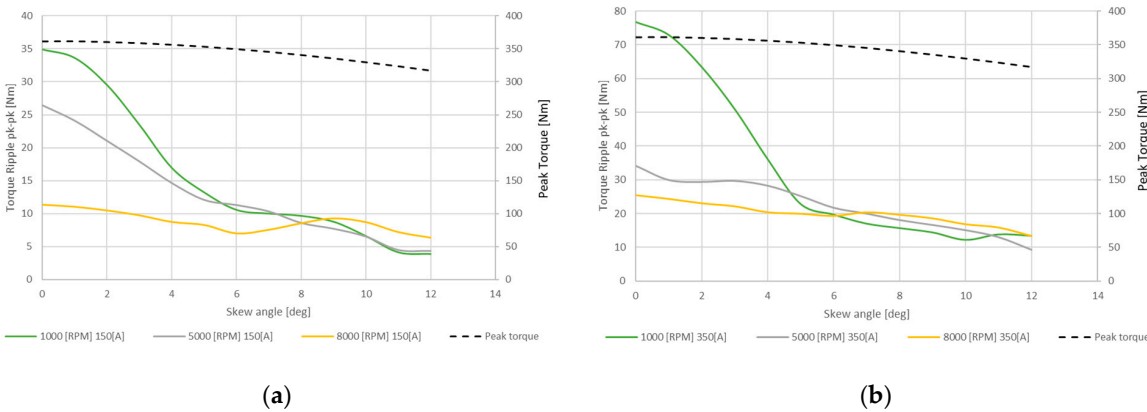

**Figure 9.** Impact of skew angles on the total torque ripple of the motor using a 4-step rotor skew: (**a**) 150 A; (**b**) 350 A.

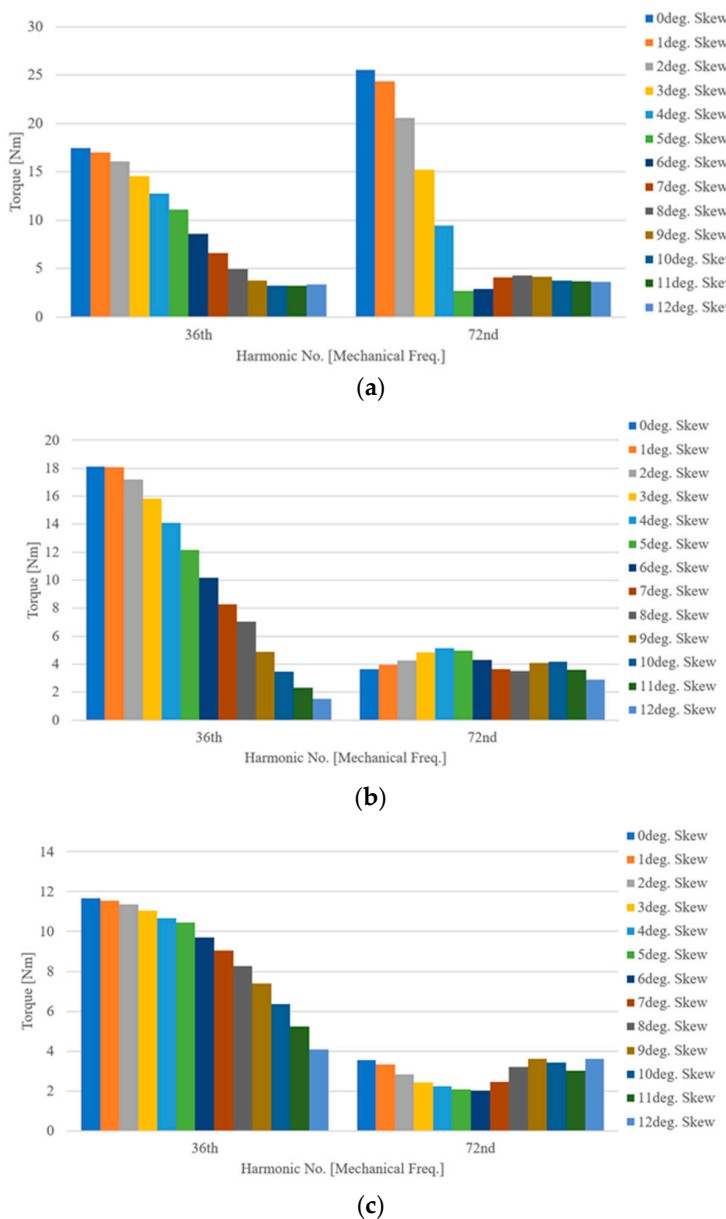

**Figure 10.** Impact of skew angles on the 36th and 72nd harmonics of torque ripple of the motor using a 4-step rotor skew for 350 A: (**a**) 1000 rpm; (**b**) 5000 rpm; (**c**) 8000 rpm.

### 3.4. Impact of Skew on Back-EMF Harmonics

Reducing torque ripple is often not the only consideration when deciding to employ skew. The harmonic content in the back-EMF of the motor can lead to a high current ripple if not compensated for in the motor controller. Table 3 shows the impact of different skewing methods on back-EMF harmonics. In this case, a 9.7% harmonic distortion is likely to cause a significant current ripple. A two-step skewing by 5° to offset the 72nd harmonic can reduce distortion to less than 5%, which is sufficient to reduce controller-induced current ripple. A 10° stator skew can give the closest waveform to a sinusoidal back-EMF.

**Table 3.** Back-EMF harmonics for different skewing methods.

| Skew Method | Harmonic Distortion Back-EMF Line-Line Voltage (%) |
| --- | --- |
| Non-skewed | 9.7 |
| 5° rotor skew (2 slices) | 4.5 |
| 5° rotor skew (4 slices) | 4.1 |
| 5° stator skew | 4.0 |
| 10° stator skew | 1.3 |

## 4. Three-Dimensional Finite Element Modeling of Axial Forces in IPM Rotors

A transient 3D finite element model of the motor was built in Ansys Maxwell as shown in Figure 11 to calculate the axial forces from the different axial stack orientations. The model only simulates an angular section of one pole of the motor to minimize its size. The key limitation of the model, however, is that it uses an isotropic material BH curve which assumes the same permeability in the axial direction (along the stack length) as compared to that along the lamination plane. This can lead to a larger field in the axial direction in comparison to an anisotropic material which is a better representation of a laminated stack.

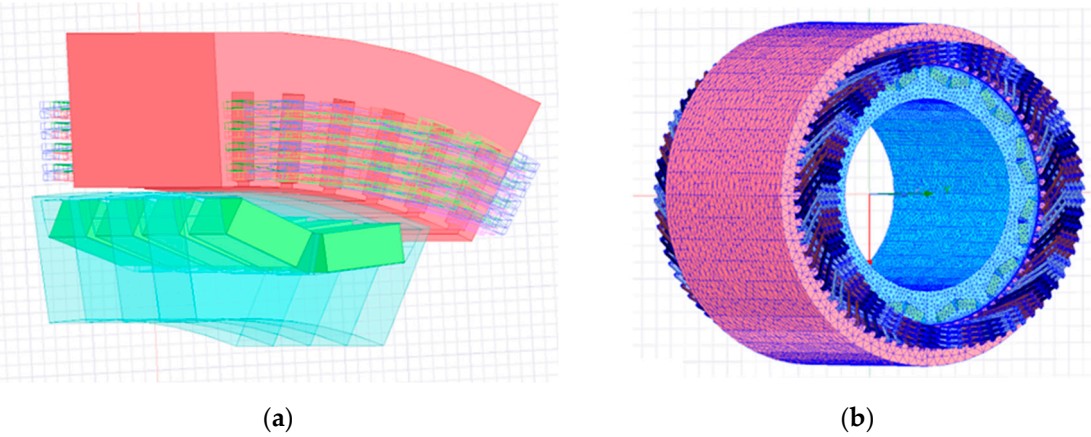

(**a**)                                                        (**b**)

**Figure 11.** (**a**) A 3D Ansys Maxwell model of a single pole section showing skewing pattern. (**b**) Mesh elements used for axial force calculation.

The different options for the skewing are shown in Figures 12 and 13, with the former showing the linear and full V configurations and the latter depicting partial V-skewing patterns. For the three-step skew, only one arrangement exists where two of the adjacent pieces are swapped axially. For the four-step skew, three arrangements can be generated as was shown. Increasing the number of steps further allows for several such combinations.

The model was run with the different skew arrangements shown in Figures 12 and 13 for six operating points within the torque speed envelope of the motor in both forward and reverse rotating directions. Figure 14 shows the impact of the different axial stack orientations in the forward direction for 150 A and 350 A. For the same phase current, the axial force decreases significantly with speed as the amount of field weakening increases. The results show that the partial V-skew arrangements can significantly reduce the axial forces, with option 3 having the largest reduction at higher speeds.

Figure 15 depicts the influence of rotational direction when using the partial V-skew at 150 A and 350 A. When comparing the forward and reverse simulations, it can be seen that direction has a small impact on force when a linear skew approach is utilized. However, when a partial V-skew (option 3) is applied, the axial force becomes more sensitive to the rotating direction [10]. This phenomenon is particularly pronounced under high current and high speed, where a substantial level of field-weakening current is applied.

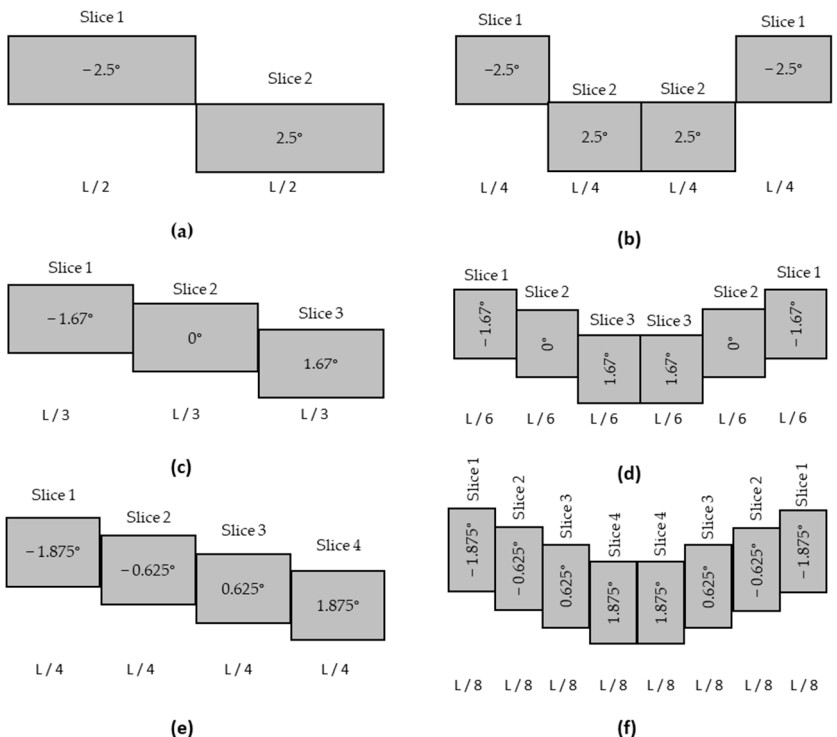

**Figure 12.** Stepwise skew arrangement for 2, 3 and 4 steps with linear or V-skew: (**a**) 2-slice linear skew; (**b**) 2-slice V-skew; (**c**) 3-slice linear skew; (**d**) 3-slice V-skew; (**e**) 4-slice linear skew; (**f**) 4-slice V-skew.

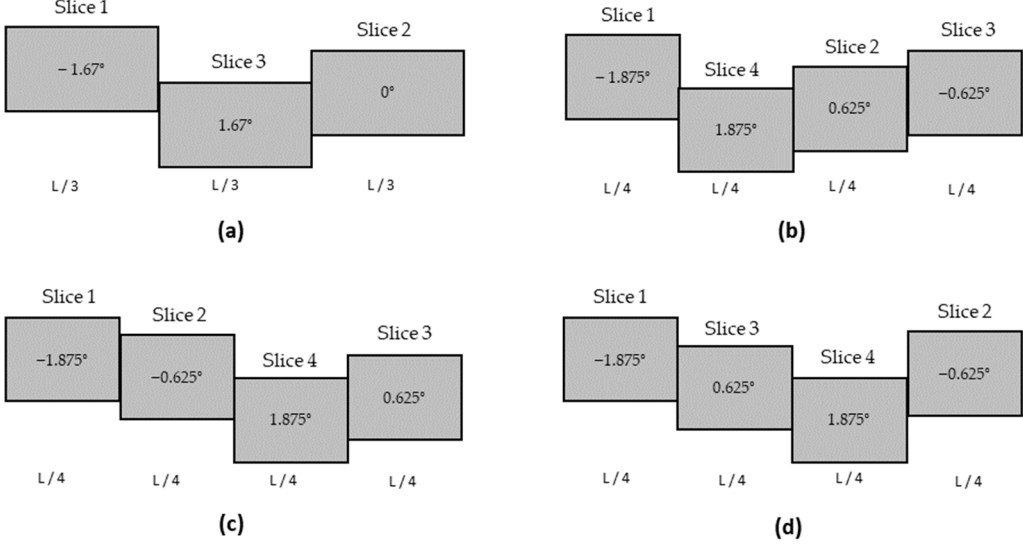

**Figure 13.** Partial V-skew options for 3 and 4 steps: (**a**) 3-slice partial V-skew; (**b**) 4-slice partial V-skew option 1; (**c**) 4-slice partial V-skew option 2; (**d**) 4-slice partial V-skew option 3.

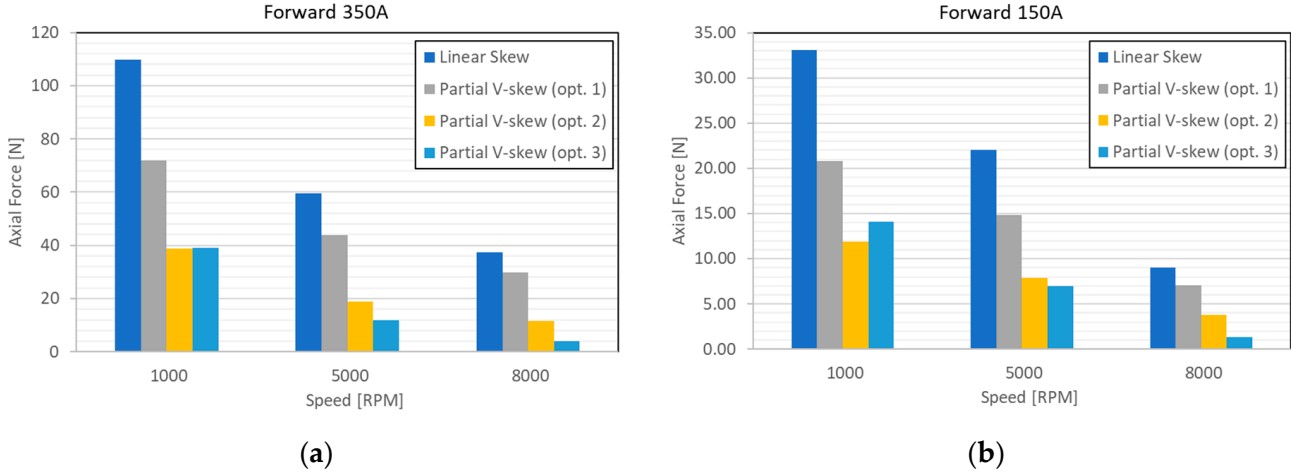

**Figure 14.** Axial forces in forward direction at different rotor speeds: (**a**) 350 A; (**b**) 150 A.

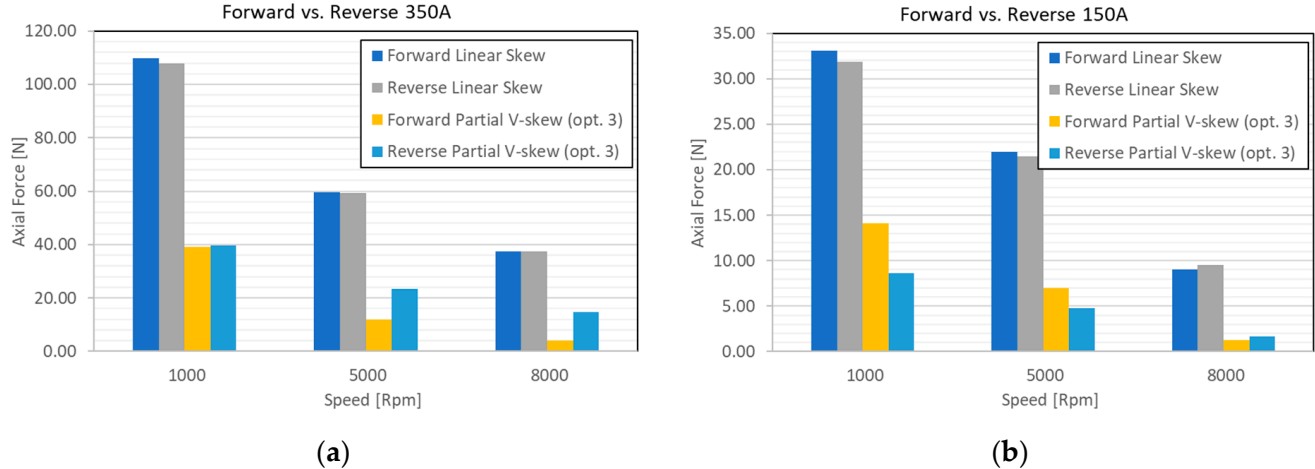

**Figure 15.** Axial forces in forward vs. reverse direction at different speeds using linear and partial V-skew: (**a**) 350 A; (**b**) 150 A.

Table 4 compares the total axial force results obtained with the analytical method presented in Section 2.3 with those via 3D FEA simulations. This comparison is performed for a phase current of 150 A and shaft speed of 1000 rpm. It can be seen that even though the partial V-skew option 3 shows a difference of almost 16% between the two methodologies, the analytical method makes an acceptable prediction of axial forces. The advantage it presents is the chance of having a predicted axial force value at an early stage, where no 3D models might be available yet. This predicted value is useful for exciting a structural model of the powertrain, as presented next.

**Table 4.** Analytical and 3D FEA total axial force results for 150 A phase current and 1000 rpm.

| | Total Axial Force (N) | | | |
|---|---|---|---|---|
| | Linear Skew | Partial V-Skew (Opt. 1) | Partial V-Skew (Opt. 2) | Partial V-Skew (Opt. 3) |
| Analytical Approximation | −29.9 | −19.9 | −9.9 | 10.0 |
| 3D FEA Simulation | −31.9 | −19.7 | −9.1 | 8.6 |
| Error | 6.4% | −1.0% | −9.1% | −15.7% |

## 5. NVH Structural Analysis

To comprehensively assess the influence of rotor skew on noise radiation, a high-fidelity structural model of the integrated powertrain was developed in the Romax Nexus suite. The structural model comprised the entire vehicle powertrain including a dual-speed gearbox, motor and housing and the stator which was interference-fit into a cooling jacket. The semi-analytical/FEA approach of Romax was used, wherein analytical models were used for calculating the stiffness matrices for gears, shafts and bearings and finite element analysis was used for the stator cooling jacket and housing. The mass contribution of stator windings was modeled by density modification of the stator tooth to simplify the model. The influence of winding stiffness could not be included as it required FEA modeling of each individual winding. In addition, 2% global damping was applied to the entire model [16], and the housing was connected to the ground through vehicle mount stiffness to ensure proper compliance.

To minimize computational effort, the dynamic model was confined to a frequency range of 10 kHz. This range facilitated effective analysis, covering up to 5 kHz in line with the Nyquist theorem. Extending dynamic simulations beyond the 10 kHz threshold requires a comprehensively fine-tuned and validated integration of both sub-structural and full structural models [5]. Due to this limitation of the maximum frequency of the condensed dynamic model, the harmonic orders of interest which were 36th and 72nd could only be evaluated at maximum operating speeds of 8000 rpm and 4000 rpm, respectively.

The dynamic model was excited with the torque ripple and radial force outputs obtained from Motor-CAD simulations in Section 3 for both the skewed and non-skewed configurations. For this case study, a 5° skew with four sections and the partial V option 3 configuration was used. The results for the mean squared velocity (MsVel) are depicted in Figure 16. The plots demonstrated a substantial reduction in MsVel attributed to the skewed rotor, particularly pronounced in the 72nd harmonic in comparison to the 36th harmonic. This reduction translates to surface velocities being lower by up to 2 orders of magnitude. The consequential damping of the surface velocities aligns with a proportional mitigation of the noise radiation associated with the 72nd harmonic component, which is notably more pronounced at lower operational speeds.

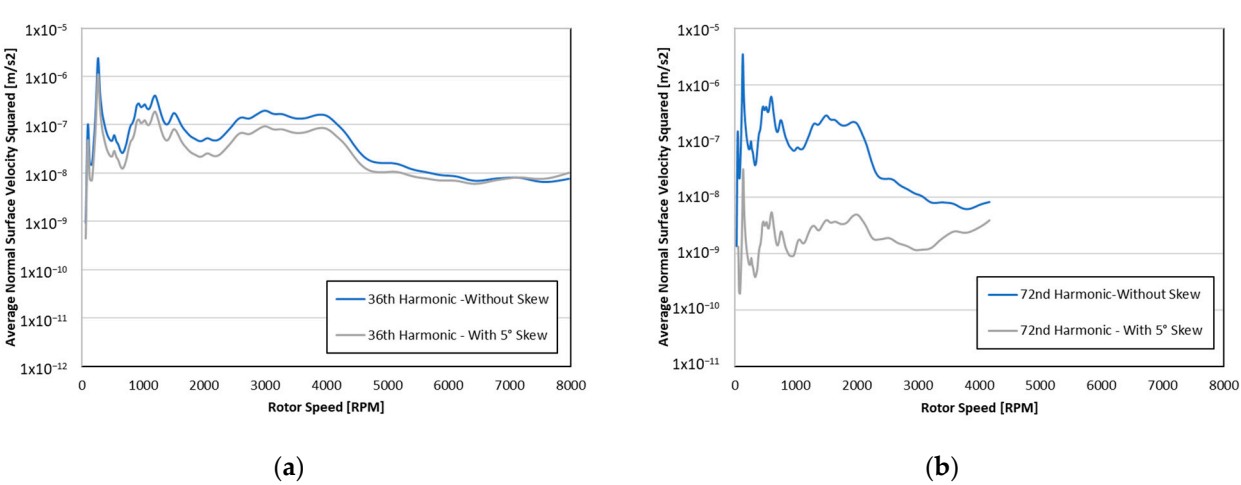

(**a**)　　　　　　　　　　　　　　　　　　　　　　　　　(**b**)

**Figure 16.** Average normal surface normal mean squared velocity with and without skew: (**a**) 36th-order excitation; (**b**) 72nd-order excitation.

## 6. Impact of Skew on Rotor Manufacture

In the motor considered in this study, a configuration of 12 poles and 2 magnets per pole was used where there are a total of 24 magnets to be inserted per stack, and the insertion time increases linearly with the number of stacks. Additionally, other production stages such as magnet adhesive dispensing, pole magnetization and magnetic measurement control will also necessitate correspondingly extended durations.

From a material perspective, there also drawbacks associated with increasing the number of stacks; considering that each magnet possesses a coating of 10–20 μm thickness on each side, a higher quantity of magnets in the axial direction leads to a reduction in the total available mass of magnetic material. As a result of manufacturing tolerances and the necessity for magnets not to extend beyond the stacks, a safety clearance of 100 μm per stack must be maintained. This clearance, in turn, contributes to a further reduction in the overall quantity of magnet mass for a given axial length. These two considerations are illustrated in Table 5, wherein the percentage reduction in magnetic mass, relative to a fixed axial length, is displayed for varying stack quantities. Notably, these factors contribute to additional reductions in average torque, which should be considered alongside the prior findings presented in Figure 8.

**Table 5.** Number of rotor stacks and its impact on active magnet material and manufacturing complexity.

| Number of Stacks | Magnet Insertions | Equivalent Axial Length Reduction |
|:---:|:---:|:---:|
| ( ) | ( ) | (%) |
| 2 | 48 | 0.19 |
| 3 | 72 | 0.29 |
| 4 | 96 | 0.39 |
| 6 | 144 | 0.58 |
| 8 | 192 | 0.77 |

## 7. NVH Optimization Workflow

Based on the analysis shown in Sections 2–6, a simplified workflow was created to enable a designer to select optimum skew parameters for their system and is shown in Figure 17. First, an analytical or 2D finite element model is used without skew, and the torque ripple harmonics are extracted over the entire operating range of the powertrain. A structural analysis is then carried out to identify operating points where response peaks are observed. A 2D multi-slice model is then used to identify the optimum skew angles for these specific operating conditions, whilst minimizing impact on average torque and considering back-EMF harmonic distortion. Based on the axial length of the motor and manufacturing constraints, the maximum number of rotor stacks is identified. The 3D FEA model is then used to study different orientations to minimize axial forces. Finally, the NVH response is recalculated to assess the reduction in NVH response and compare it with set targets.

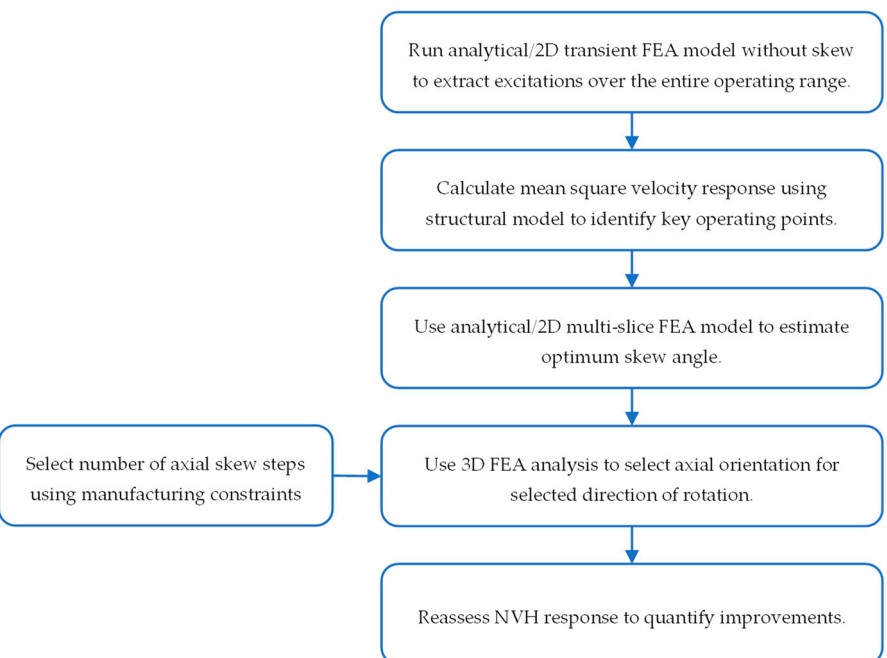

**Figure 17.** NVH optimization process for skewed IPM rotors.

## 8. Conclusions

This paper presents a comprehensive study of the impact of rotor skew on the performance of IPM motors. Detailed mathematical models are proposed for torque ripple and cogging torque which can accurately predict the magnitudes of the torsional harmonics and the impact from skew. Additionally, a simplified analytical framework is presented to derive the total axial force in IPM rotors. This model is formulated as a function of main parameters such as the stator MMF, the average airgap flux density and the specific skew angles between slices. The analytical models of torque ripple and cogging are assessed using Ansys-MotorCAD (v2022.1.1.1. Wrexham, UK) and the outcomes are compared for an automotive traction motor. The comparison shows a satisfactory comparison for both characteristic frequencies and amplitudes.

The study also investigates the influence of the skewing angle using a real traction motor example. For the motor used in this paper, a 10° skewing angle proves effective in minimizing all harmonics for cogging and ripple. However, this approach entails a trade-off, leading to a large decrease in peak torque. Alternatively, acceptable results over a wide range of operating points can be achieved with a lower angle between 5° and 6°, mitigating the impact on peak torque. Care was taken to ensure that no peaks in the NVH response for the 72nd harmonic were observed at 5000 rpm as skewing increased the excitation for this condition. The impact of skew on back-EMF harmonics was also evaluated, underscoring the need for a balanced approach to enhance both torque quality and motor control.

Once the number of slices and total skewing angle were selected, the significance of the axial order in which they are placed and their effect on total axial force was assessed in Section 4. A 3D FEA model was built using another commercial tool to obtain the axial forces for a four-slice rotor when considering different skewing patterns such as linear, full V and three different options for a partial V. The rotating direction—forward or reverse—is presented as a minor variable but is important for traction motors where the vast majority of the time the motor rotates in a fixed direction. Results are compared between analytical prediction and 3D FEA, showing the correct trends and acceptable error, as these axial forces are mainly used to obtain an order of magnitude for the NVH excitations rather than a specific value.

A high-fidelity structural model of the integrated powertrain was established using commercial software that employs a semi-analytical/FEA approach to compute the stiffness matrices of various drivetrain components. The model was limited to a maximum frequency of 10 kHz and was excited with torque ripple and axial forces derived from the preceding sections, accounting for the selected number of stacks and skewing angle. Results for MsVel reveal a substantial reduction in the 72nd harmonic compared to the 36th, translating into surface velocities lowered by up to 2 orders of magnitude. Noise radiation is notably mitigated, particularly at lower speeds where the 72nd harmonic prevails.

A manufacturing analysis was conducted, considering the number of slices and, consequently, the required magnet insertions. An increased number of slices results in more rotor interfaces, leading to a reduction in the overall amount of magnetic material in the rotor due to tolerance stack-up analysis, consequently also affecting peak torque. Finally, a straightforward yet practical workflow was devised to allow designers to select the optimal skew parameters.

**Author Contributions:** Conceptualization, methodology, visualization, validation and supervision, L.K.R. and J.M.G.; software, L.K.R., J.M.G., T.C., A.H. and V.S.; writing—original draft preparation and writing—review and editing, L.K.R., J.M.G. and A.H. All authors have read and agreed to the published version of the manuscript.

**Funding:** The work conducted was funded entirely by ARRIVAL SA.

**Data Availability Statement:** Data are contained within the article.

**Conflicts of Interest:** The authors declare no conflict of interest. A.H. is an employee of ARRIVAL SA. The paper reflects the views of the scientists, and not the company.

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
