# Peer review of "A Methodology for Applying Skew in an Automotive Interior Permanent Magnet Rotor for Robust Electromagnetic and Noise, Vibration and Harshness Performance"

_wevj, doi:10.3390/wevj14120350_

Round 1

Reviewer 1 Report

Comments and Suggestions for Authors

The authors present a technique to mitigate torque ripple and NVH in IPM machines used in automotive applications. I reviewed the paper and have the following comments:

- The authors should have been more careful about the status of the paper before submission. The paper is full of missing references where the phrase "Error! Reference source not found" is all over the paper.

- My biggest concern is that the paper shows no mathematical models, and it relies completely on simulation. Scientific research shall be supported with mathematical models. Simulators are tools to be utilized to conduct research.

I recommend that the authors to reconstruct this paper and support their research work with mathematical models relevant to the IPM performance.  

Comments on the Quality of English Language

English language is adequate.

Author Response

Uploaded as a file 

Reviewer 2 Report

Comments and Suggestions for Authors

In the submission, the skewing strategy for a typical traction motor using 2D and 3D FEM is developed, and the optimal number of axial stacks is defined to obtain a convenient trade-off between reducing axial forces and increasing manufacturing costs. The detailed comments are listed as follows,

1, the acronym should be defined, for example, the term ‘NVH’ in the abstract.

2, the citations in the manuscript should be updated, for example, the citation on line 92 of page 3 is missing.

3, the figures about the result of the NVH and FEA model should be provided.

4, for the figure 6, is there any real slice? Please add more details.

5, A transient 3D finite element model in Ansys Maxwell should be added.

6, in Figure 9 (b), the rotating speed is limited to 4000 rpm, but the rotating speed in Figure. 9 (a) is about 8000 rpm, please add more explanations.

7, is there any result about the FEA analysis using the ANSYS MAXWELL software?

8, more comparisons should be conducted to verify the effectiveness of the proposed analysis method.

Comments on the Quality of English Language

Moderate editing of English language required

Author Response

Uploaded as a file 

Round 2

Reviewer 1 Report

Comments and Suggestions for Authors

I am fine with this final version.

Reviewer 2 Report

Comments and Suggestions for Authors

no more comments, all problems are addressed well.

Comments on the Quality of English Language

 Minor editing of English language required